# Altered Expression of NK Receptors in Racially/Ethnically Diverse and Risk-of-Relapse Pediatric Acute Lymphoblastic Leukemia Patients

**DOI:** 10.3390/biomedicines13061412

**Published:** 2025-06-09

**Authors:** Stephen Mathew, Roslin Jose George, Alexsis Garcia, Sheila Powers, Subhash Aryal, W. Paul Bowman

**Affiliations:** 1Department of Microbiology, Immunology and Genetics, University of North Texas Health Science Center, Fort Worth, TX 76107, USA; alexsisgarcia@my.unthsc.edu (A.G.); sheila_powers@yahoo.com (S.P.); 2Mayo Clinic Hospital, Rochester, MN 55902, USA; george.roslinjose@mayo.edu; 3School of Nursing, Johns Hopkins University, Baltimore, MD 21205, USA; saryal5@jh.edu; 4Cook Children’s Medical Center, Fort Worth, TX 76104, USA; paul.bowman@cookchildrens.org

**Keywords:** acute lymphoblastic leukemia (ALL), health disparities, risk factors, natural killer (NK) cell, immune receptors

## Abstract

**Background/Objectives:** Acute Lymphoblastic Leukemia (ALL) is a cancer that predominantly affects white blood cells within the blood and bone marrow of adults and children. Currently, ALL is one of the most prevalent malignancies in pediatric patients and is most seen among Caucasian and Hispanic descent, with lower incidence in African American children. The goal of the study was to investigate the expression of immune cell receptors in racial/ethnic populations and risk factors for relapse that could potentially influence the pediatric ALL outcomes. **Methods:** Twenty healthy subjects and forty-two pediatric ALL subjects were enrolled in the study and whole-blood was collected at diagnosis and post-chemotherapy, and the cell surface expression of various immune receptors, including 2B4, CS1, LLT1, Nkp30, and NKp46, was determined by flow cytometry. **Results:** Very high-risk and high-risk of relapse ALL subjects showed increased expression of LLT1 on NK cells, T cells, and monocytes at diagnosis compared to healthy subjects. CS1 was also significantly overexpressed on monocytes of very-high risk ALL subjects both at diagnosis and after the end of chemotherapy as compared to healthy subjects. Also, there was a significantly increased expression of NKp30 on T cells of Caucasians as compared to Hispanics and African Americans at diagnosis, and downregulation of CS1 and LLT1 on T cells of Caucasians post-induction chemotherapy. **Conclusions:** The altered expression of immune receptors in racial/ethnic and risk stratified groups may provide insights into the immune surveillance mediated by T cells and NK cells against pediatric ALL.

## 1. Introduction

Acute lymphoblastic leukemia (ALL) is marked by the formation of abnormal lymphoblasts, leading to a broad range of symptoms. While both children and adults can present with ALL, ALL remains the most prevalent childhood cancer in children under the age of 15, with incidence rates being higher in boys than in girls [1,2,3,4,5]. Predominantly, individuals of Caucasian and Hispanic descent present with ALL while African American children have relatively lower incidence [6,7,8,9]. Patients with ALL display symptoms related to bone marrow failure such as fatigue, pallor, and fever along with easy bruising and bleeding [10,11,12,13]. Currently, first-line treatments include vincristine, glucocorticoids, and doxorubicin along with allogenic stem cell transplantation [2,3,9,14,15]. Despite the advances in various treatments, approximately 20% of pediatric patients experience a relapse [16] and of those treated with chemotherapeutic agents roughly 80–90% remained in remission [3,17,18,19,20,21].

Previously, age and white blood cell count were primarily used to risk stratify patients, with worse prognosis being associated with increased age [11]. Currently, various factors are used to assess risk stratification which include genetic information as well as age, white blood cell count, minimal residual disease (MRD) response, time of relapse from diagnosis, site of relapse, and immunophenotype (B-ALL or T-ALL) [4,11,22,23]. In addition to risk stratification, there have been various studies assessing the correlation between racial/ethnic health disparities and ALL [7,19,24,25,26]. Previous studies have noted that while African Americans have a lower incidence rate in comparison to their white and non-white counterparts, they often have worse treatment outcomes and lower rates of survival [6,27,28]. In addition, a link between genetic variations in African American and Hispanic pediatric patients and ALL risk/susceptibility have previously been investigated [24,28]. It has been noted that African Americans are more likely to have T-ALL, the more aggressive ALL lineage compared to B-ALL [29].

Some studies have investigated a link between ALL and natural killer (NK) cell dysfunction [20,30,31,32]. NK cells play a vital role in the innate immune system and are the body’s first line of defense against foreign invaders and cancer cells [33,34]. Numerous recent studies have shed light on various membrane-bound receptors involved in the recognition, activation, and inhibition of NK cells through interactions with other immune cells [35,36]. Recent clinical studies have shown that chimeric antigen receptor (CAR)-T cells and CAR-NK cells are efficiently able to reduce relapses of ALL and more studies need to be conducted to fully realize the potential of this immunotherapy [37,38]. In this study, we have focused on five receptors/ligands, namely 2B4 (CD244, SLAMF4), CS1 (CD319, SLAMF7), LLT1 (CLEC2D), NKp30 (NCR3), and NKp46 (NCR1). Previously, our group had cloned 2B4, CS1, and LLT1 and evaluated their expression and function in various cancers including prostate cancer, triple-negative breast cancer, and acute lymphoblastic leukemia (ALL) [20,31,39,40]. These receptors have been linked to NK cell dysfunction via the disruption of optimal immune surveillance in ALL patients and various diseases [21,41,42,43,44]. Data collected from ALL subjects (ages 2–19, B-ALL and T-ALL) both at day 0 (diagnosis) and day 29 (post-chemotherapy treatment) were analyzed with a focus on the immune receptor expression in the following racial/ethnic populations: African Americans, Hispanics, and Caucasians. We also analyzed the expression of these receptors/ligands in risk-of-relapse stratified groups pre- and post-chemotherapeutic treatments. Overall, this study offers fresh insight into the dysregulation of immune receptor/ligand expression in ALL patients among various racial/ethnic and risk-stratified populations.

## 2. Materials and Methods

### 2.1. ALL and Healthy Subjects

Pediatric patients aged 2 to 21 years with newly diagnosed ALL were enrolled in the study at the Hematology and Oncology Clinic of Cook Children’s Medical Center (CCMC) in Fort Worth, TX, USA. Informed consent/assent was obtained from each subject by Dr. W. Paul Bowman, MD or nursing staff per IRB approval from the University of North Texas Health Science Center (UNTHSC), Fort Worth, TX, USA and CCMC (UNTHSC IRB# 2008-094 & CCMC IRB#2008-57). Additionally, children under 21 years of age who visited the pediatric clinic at the UNTHSC Health Pavilion for routine medical check-ups were enrolled as healthy controls. Disease profile and demographic information were collected for every subject.

### 2.2. Blood Collection

One blood sample (8 mLs) indicated as the first blood draw (1BD) was collected by the staff from every ALL subject after obtaining written informed consent before any treatment was provided to them. A second blood sample indicated as the second blood draw (2BD) was collected on day 29 at the end of chemotherapy treatment, which typically lasts for 4 weeks. Healthy subjects only provided a single blood sample. A total of 42 ALL subjects and 20 healthy subjects were enrolled in the study.

### 2.3. Isolation of Peripheral Blood Mononuclear Cells (PBMC)

Peripheral blood mononuclear cells (PBMCs) were isolated from whole-blood collected in purple-top tubes containing ethylenediamine tetra acetic acid (EDTA) by Histopaque-1077 (Sigma Chemicals, St Louis, MO, USA) density gradient centrifugation using LeucoSep tubes (Greiner, Monroe, NC, USA). Red blood cells were lysed using ACK lysis buffer. The isolated PBMCs were subsequently utilized for flow cytometric analysis of immune receptor expression.

### 2.4. Antibodies and Immunostaining for Flow Cytometry

Prior to immunostaining, PBMCs were incubated with human IgG Fc fragments (Rockland Immunochemicals, Philadelphia, PA, USA) to block potential Fc receptor-mediated fluorescence. The antibodies used to distinguish immune cell populations included FITC-conjugated anti-human CD3 monoclonal antibody (mAb) for T cells, PE-Texas Red-conjugated anti-human CD19 mAb for B cells, APC-conjugated anti-human CD56 mAb for NK cells, and APC-Cy7-conjugated CD14 mAb for monocytes. To assess the percentage and quantitative cell surface expression of each immune receptor and its ligands, PE-conjugated monoclonal antibodies against human 2B4, CS1, LLT1, NKp30, and NKp46 were used for labeling. Immune cell populations were gated based on forward and side scatter to (Appendix A) distinguish lymphocytes and monocytes from other cells according to size and granularity, using a Beckman Coulter FC500 flow cytometer (Beckman Coulter Life Sciences, Indianapolis, IN, USA). All antibodies used in the study were obtained from BioLegend, San Diego, CA, USA. The median fluorescence intensity ratio (MFIR) was calculated by comparing the median fluorescence intensity (MFI) of test samples to that of isotype control samples. MFIR for each sample was determined by dividing the MFI of the test sample by the MFI of the corresponding isotype control.

### 2.5. Statistical Analysis

A multivariate analysis of variance (MANOVA) was conducted to evaluate differences in immune receptor expression among healthy individuals and ALL patient samples at diagnosis and after chemotherapy, across various racial/ethnic groups and risk stratification categories, using SAS version 9.4 (SAS, Cary, NC, USA). The statistical test adjusts for correlations among multiple measurements taken from the same subject. When a statistically significant overall difference was identified using MANOVA, each variable was individually assessed by race/ethnicity and risk-stratified groups using the ANOVA method. Significance was determined at alpha = 0.05. Graphs were created by using Graph Pad Prism version 10.2.3 software (GraphPad, San Diego, CA, USA).

## 3. Results

### 3.1. Altered Cell Surface Expression of CS1, LLT1 and NKp30 in CD3+ T Cells of Hispanic, Caucasian and African American ALL Subjects Pre- and Post-Chemotherapy Treatments

PBMCs were isolated from the whole blood of 42 ALL patients and 20 healthy individuals and the cell surface expressions of 2B4, CS1, LLT1, NKp30, and NKp46 were analyzed. CS1 and LLT1 expression on CD3+ T cells of Caucasians was significantly downregulated as compared to Hispanics and African Americans post-chemotherapy (2BD), whereas at diagnosis (1BD) there was no significant difference in the expression of those receptors between the racial/ethnic populations (Figure 1C–F,K,L). As shown in Figure 1M, increased cell surface expression of NKp30 was observed on the CD3+ T cell population of Caucasians (36.2% positive cells) as compared to Hispanics (16.7% positive cells) and African Americans (8.4% positive cells) at diagnosis (*p* < 0.01; *p* < 0.001). However, following the completion of chemotherapy (2BD), NKp30 expression in Caucasians decreased to levels comparable to those observed in Hispanic and African American individuals (Figure 1N). There was no significant difference in the expression of 2B4, NKp30 (MFIR) and NKp46 among the different racial/ethnic populations (Figure 1A,B,G–J).

### 3.2. Increased Cell SurfaceEexpressions of LLT1 and NKp30 and Downregulation of CS1 in CD3+ T Cells of High-Risk ALL Subjects

ALL subjects were stratified into risk-of-relapse groups and were designated as very high-risk (VHR), high-risk (HR), or standard risk/low-risk (SR-LR) and the cell surface immune receptor expression on CD3+ T cells was determined. The risk stratification was based on the subject’s MRD status, age, white blood cell count, CNS status, DNA index, ALL subtype, hyperdiploidy, hypodiploidy, TEL/AML status, MLL translocations, Philadelphia (Ph) chromosome, t(1;19) translocation, and additional chromosomal abnormalities. The immune receptor expression was determined in both healthy (only one sample) and ALL subjects at diagnosis (day 0, 1BD) and post-induction chemotherapy (day 29, 2BD). As shown in Figure 2E, LLT1 was observed to be significantly overexpressed in CD3+ T cells of high-risk ALL (31.67%) subjects as compared to healthy (5.8%) subjects (*p* < 0.01) at diagnosis (day 0, 1BD), which dropped down post-induction chemotherapy (day 29, 2BD) treatment to comparable levels of expression (12.17%) with healthy subjects (Figure 2F). Similarly, cell surface expression of NKp30 was significantly upregulated on T cells of high-risk patients (40.55%) at diagnosis (1BD) as compared to standard risk/low-risk and healthy (13.4%) subjects (Figure 2G). NKp30 expression on T cells in high-risk subjects decreased to 27.3% following chemotherapy (2BD) treatment (Figure 2H). In contrast, CS1 expression on CD3+ T cells, measured by median fluorescence intensity ratio (MFIR), was significantly reduced at diagnosis (day 0, 1BD) in both very high-risk (1.76) and high-risk (2.47) ALL patients compared to healthy (3.89) individuals (Figure 2K). After induction chemotherapy (day 29, 2BD) there was no significant difference in CS1 expression on T cells in the different risk-stratified groups (very high-risk = 2.7, high-risk = 2.09, and standard risk/low-risk = 2.33) of ALL subjects as compared to healthy (3.89) subjects (Figure 2L). There was no significant difference in the percentage of positive cell expression of 2B4, CS1 and NKp46 on CD3+ T cells in the different risk-stratified groups of ALL subjects as compared to healthy subjects either at diagnosis (1BD) or post-induction chemotherapy (2BD) treatment (Figure 2A–D,I,J).

### 3.3. Increased Cell Surface Expression of LLT1 Was Observed in CD56+ NK Cells of Very High-Risk and Standard/Low-Risk ALL Patients

LLT1 cell surface expression was significantly elevated in CD56+ NK cells of very high-risk (49.91% and 49.77%) and standard risk/low-risk (34.64% and 31.38%) ALL subjects at diagnosis (1BD) and after chemotherapy (2BD), respectively, as compared to healthy (12.64%) subjects (Figure 3E,F). CD56+ NK cells from very high-risk ALL patients at diagnosis showed elevated expression of CS1 (61.34%), NKp30 (74.65%), and NKp46 (68.17%) as compared to healthy subjects (39.84%, 46.1%, and 49.9%, respectively), but it was not statistically significant (Figure 3C,G,I).

A similar trend of high expression of CS1 (67.88%), NKp30 (65.44%), and NKp46 (57.79%) was observed in the CD56+ NK cells of very high-risk ALL subjects post-induction (2BD) chemotherapy as compared to healthy subjects (39.84%, 46.1%, and 49.9%, respectively), but it was not statistically significant (Figure 3D,H,J). Contrastingly, a decrease in the expression of 2B4 on CD56+ NK cells was observed in the high-risk (72.16%) and standard risk/low-risk (60.2%) ALL subjects at diagnosis (1BD) as compared to healthy (77.04%) subjects (Figure 3A). After chemotherapy treatments (day 29, 2BD), there was a further decline in 2B4 expression on NK cells in the different risk-stratified groups (very high-risk = 71.6%, high-risk = 64.51%, and standard risk/low-risk = 55%) of ALL patients as compared to healthy (77.04%) individuals (Figure 3B).

### 3.4. Enhanced Cell Surface Expression of CS1 and LLT1 in CD14+ Monocytes of Very High-Risk and High-Risk ALL Subjects

CS1 cell surface expression was observed to be significantly higher in CD14+ monocytes of very high-risk (39.32%; *p* < 0.05) and high-risk (38.08%; *p* < 0.01) ALL subjects, respectively, at diagnosis (1BD) as compared to healthy (14.71%) subjects (Figure 4C). The standard risk/low-risk ALL subjects also showed high expression of CS1 (29.77%) in the monocyte population at diagnosis (1BD). The increased expression of CS1 on monocytes remained unchanged even after induction chemotherapy (2BD) in very high-risk (41.01%, *p* < 0.05) and standard risk/low-risk (36%, *p* < 0.05) and was significantly higher compared to healthy (14.71%) subjects (Figure 4D). High-risk and standard risk/low-risk ALL subjects showed significantly higher expression of 73.61% (*p* < 0.05) and 69.71% (*p* < 0.05) positive cells for LLT1 on CD14+ monocytes at diagnosis (1BD) as compared to healthy (44.9%) subjects (Figure 4E). The expression of LLT1 on monocytes decreased after induction chemotherapy (2BD) in very high-risk (45.23%), high-risk (60.8%), and standard risk/low-risk (47.29%) ALL subjects but was higher than the healthy (44.9%) subjects (Figure 4F). There was no significant difference in 2B4, NKp30, and NKp46 expression on CD14+ monocytes in the different risk-stratified groups of ALL patients as compared to healthy individuals either at diagnosis (1BD) or post-induction chemotherapy (2BD) treatment (Figure 4A,B,G–J).

### 3.5. Clinicopathological Features of ALL Subjects

Forty-six ALL patients and twenty healthy individuals were enrolled in this study. Four patients failed to satisfy the inclusion criteria or failed to complete the study and so were excluded from data analysis. There were equal number of male (*n* = 21) and female subjects (*n* = 21). A total of 93% of the subjects had B-ALL whereas 7% had T-ALL (Table 1 and Appendix A). We also enrolled twenty healthy subjects that we attempted to match with the patient groups regarding age, gender and race/ethnicity, but these were not perfectly matched due to the complexity of enrolling healthy pediatric subjects into the study.

The clinicopathological features revealed that 14.3% (*n* = 6) had a positive MRD status, whereas 81% (*n* = 34) had a negative MRD status and 5% (*n* = 2) were unknown. Analyzing the CNS status revealed that 76.2% (*n* = 32) of ALL subjects did not show any blasts in the cerebrospinal fluid (CSF) at diagnosis, with 12% (*n* = 5) showing CNS status 2a, 1 subject having CNS status 2b, and a couple of the subjects indicating CNS status 2c and 3a. A total of 24% (*n* = 10) of the ALL subjects showed a normal DNA index of 1.0, whereas 21.43% (*n* = 9) displayed an elevated DNA index (1.05–2.0) and another 21.43% (*n* = 9) reported a highly elevated DNA index (>2.0). For 33% (*n* = 14) of ALL subjects the DNA index was unknown. Among the ALL subjects, 28.5% (*n* = 12) displayed hyperdiploidy, characterized by trisomies of chromosomes 4, 10, or 17, occurring in one, two, or all three chromosomes, whereas 2.38% (*n* = 1) showed hypodiploidy. A couple of the subjects were positive for TEL/AML, which is expressed from the t(12;21) chromosomal translocation in B-precursor ALL patients and six subjects were positive for Philadelphia chromosome [(Ph)/BCR/ABL-positive].

## 4. Discussion

Various risk factors can influence the chances of relapse in ALL patients, which includes immunophenotype specifically T cell ALL which is considered to be more aggressive in comparison to B-cell ALL. Survival rates among ALL patients vary based on gender, race/ethnicity, and socioeconomic status along with immunophenotype [20,26,45]. NK cells play a vital role in the immune surveillance of cancers and viral infections. NK cell activation is managed by a suite of activating, co-stimulatory, and inhibitory receptors and the regulation of these signals ultimately determines the cell function and activation [46,47,48]. The activating receptors are not able to activate NK cells on their own and so need the synergy of several receptors. The ITAMs (immunoreceptor tyrosine-based activation motifs) initiate activating signals when phosphorylated, leading to recruitment of kinases like Syk and activation of pathways such as PLCγ, PI3K, and calcium signaling. The inhibitory receptors’ interaction with their corresponding ligands causes the phosphorylation of ITIMs (immunoreceptor tyrosine-based inhibitory motifs) sequences in the inhibitory receptor mediated by the Src-family kinase (SFK) [49,50]. The ITIMs activate protein tyrosine phosphatases (PT-Pases), mainly SHP-1 and SHP-2 after phosphorylation. SHP-1 further downregulates multiple activating signal molecules by dephosphorylation, inhibiting the NK cell activation signal and eventually inhibiting NK cell activity [50]. In this study we wanted to investigate the cell surface expression of immune receptors 2B4, CS1, LLT1, NKp30, and NKp46 in different racial/ethnic populations and risk-of-relapse stratified groups. These receptors are predominantly expressed on NK cells and other immune cells and play a role in cancers and other diseases. Our study has shown the differential expression of immune receptors in different immune cells in racial/ethnic populations and risk-of-relapse stratified groups.

The majority of the subjects recruited in our study were B-precursor ALL subtype, with only a few that were T cell ALL. Our results show that there is altered expression of CS1, LLT1, and NKp30 in different racial/ethnic populations at diagnosis and induction chemotherapy further caused a downregulation in the expression of these receptors in Caucasians as compared to Hispanics and African Americans that could impact the immune dysregulation (Figure 1). Prior studies have predominantly examined genetic polymorphisms and racial/ethnic influence on the risk to develop childhood ALL that may help explain varying susceptibilities across groups to environmental toxins [51]. de Smith et al. reported that Hispanic/Latino children with acute lymphoblastic leukemia (ALL) are more likely to carry high-risk germline SNPs, such as those in the *GATA3* and *IKZF1* genes. The *GATA3* SNP rs3824662, which is more prevalent in individuals with Indigenous American ancestry, has been strongly associated with Philadelphia chromosome-like (Ph-like) ALL—a subtype known for its poor response to standard chemotherapy and increased relapse risk. Similarly, *IKZF1* variants, also enriched in Latino populations, are associated with impaired lymphoid development and reduced survival [6]. Moreover, in pediatric acute myeloid leukemia (AML) a germline SNP in the epigenetic regulator gene *TET2* (rs2454206) has been identified as a prognostic marker influencing survival outcomes. This SNP shows variable prevalence across racial groups, with the *TET2^AA^* genotype being more common in Black patients showing higher non-relapse mortality, mainly due to infection-related complications during treatment, suggesting increased treatment toxicity rather than differences in relapse rates. Functional studies indicated that rs2454206 affects the expression of *CXXC4*, a negative regulator of *TET2*, potentially altering epigenetic regulation and host susceptibility to chemotherapy toxicity [52]. These studies underscore the importance of racial/ethnic differences in SNP distribution that can influence not only chemoresistance but also treatment-related complications. A study on Axicabtagene ciloleucel (axi-cel), an autologous anti-CD19 chimeric antigen receptor (CAR) T cell therapy for relapsed/refractory (R/R) large B-cell lymphoma (LBCL), showed that non-Hispanic (NH) Black patients had a lower overall response rate and a lower complete response rate than NH White patients. NH Black patients also had a shorter progression-free survival vs. NH White and NH Asian patients. They did not find any difference in cytokine-release syndrome by race/ethnicity; however, higher rates of any-grade immune effector cell-associated neurotoxicity syndrome were observed in NH White patients than in other patients [53].

Increased expression of LLT1 was observed on T cells of high-risk ALL patients at diagnosis as compared to healthy individuals. But at the end of induction chemotherapy the expression of LLT1 on T cells of high-risk ALL patients were comparable to healthy individuals (Figure 2). It is well-documented that T cell exhaustion can result in failure of CAR-T cell therapy. B-cell recovery in pediatric B-ALL within 3 months of CD19 CAR-T cell therapy indicates a high-risk of relapse potentially due to T cell exhaustion [54]. Other studies have indicated that the spontaneous aggregation of antigen by CAR-T cells causes CAR-CD3ζ cells to accumulate and leads to CAR-T cell exhaustion [55]. Research has demonstrated that LLT1 on tumor cells interacts with the NK cell inhibitory receptor NKRP1A (CD161), leading to the suppression of NK cell-mediated tumor cytotoxicity [39,40,56,57,58,59,60,61,62,63]. Given that LLT1 is upregulated in various cancers and transmits inhibitory signals to T cells, it is plausible that increased LLT1 expression on ALL cells may potentially cause T cell exhaustion and increased risk-of-relapse. Elevated LLT1 expression on T cells in high-risk ALL patients may contribute to the inhibitory LLT1–NKRP1A interaction with NK cells, potentially enabling leukemic cells to evade NK cell-mediated cytotoxicity. Moreover, on NK cells we observed an increased expression of LLT1 in the very high-risk, high-risk, and standard risk/low-risk ALL patients both before and after chemotherapy treatment as compared to the healthy individuals (Figure 3E-F), which may potentially cause chronic activation of NK cells and be an indicator for NK cell exhaustion, which is where they are unable to recognize malignant cells and carry out the cytotoxic effects [33,64,65]. Similarly, LLT1 expression on CD14+ monocytes was significantly increased in very high-risk, high-risk, and standard risk/low-risk ALL subjects in comparison to their healthy counterparts (Figure 4E), suggesting that the high expression of LLT1 in different cell types could be a major contributor to cancer prognosis and risk in ALL.

At diagnosis, CD56+ NK cells from very high-risk ALL subjects exhibited high expression of CS1, NKp30, and NKp46 as compared to healthy subjects (Figure 3C,G,I) and even after chemotherapy the expression of these receptors did not change significantly but were higher than the expression in healthy subjects (Figure 3D,H,J). Studies have shown that there is increased expression of CS1 in plasma cells, multiple myeloma, and systemic lupus erythematosus [42,43,66,67,68,69]. Chretien et al. reported acute myeloid leukemia patients with higher expression of NKp46 had better progression free survival (PFS) and overall survival (OS) than their counterparts with lower expression [44]. Other studies found that NKp30 down-regulation plays a vital role in cervical cancer and HPV-16 immune evasion [70,71]. In contrast, our findings showed high expression of NKp30 on T cells, NK cells, and monocytes of high-risk ALL subjects, which could suggest that NKp30 may play a role in the relapse of ALL. These results align with previous studies which further indicate a connection between these receptors with cancer risk and severity.

In CD14+ monocytes, very high-risk and high-risk ALL subjects showed increased cell surface expression of CS1 that was statistically significant at diagnosis (1BD) as compared to healthy subjects (Figure 4C) and chemotherapy treatment (2BD) did not change the CS1 expression in very high-risk and standard risk/low-risk ALL subjects (Figure 4D). We previously demonstrated that CS1 is expressed in activated monocytes through PI3K- and NF-κB-dependent pathways and exerts an inhibitory effect by decreasing the production of proinflammatory cytokines in LPS-stimulated monocytes [43]. Li et al. reported that CS1 expressing monocytes may promote high-risk of relapse, progression of disease and immunosuppressive state to increase T cell exhaustion and suppression of NK cell function in multiple myeloma [54]. The inhibitory role of CS1 on monocytes could potentially contribute to poorer outcomes in very high-risk and high-risk ALL subjects.

## 5. Conclusions

Collectively, the results of our study demonstrate that there is differential expression of immune receptors in different racial/ethnic and risk-of-relapse stratified groups in pediatric ALL subjects. The results also suggest that variations in receptor and ligand expression among different racial/ethnic groups and risk stratifications may influence T cell- and NK cell-mediated immune surveillance in pediatric ALL. A major limitation to our study is the small sample size, and the altered expression of receptors observed in different racial/ethnic and risk-of-relapse groups needs to be validated in a larger cohort study. Although we have focused on immune receptor/ligand expression in racial/ethnic populations, other studies have shown genetic risk factors that contribute to the differences in incidences and overall survival in ALL, but the effect of environmental factors and tumor immune microenvironments cannot be ruled out. While our study brings greater understanding to the expression of various receptors/ligands, further studies are necessary to evaluate the function of these receptors and their effect on cancer prognosis and overall survival in risk stratified ALL patients from different racial/ethnic populations.

## Figures and Tables

**Figure 1 biomedicines-13-01412-f001:**
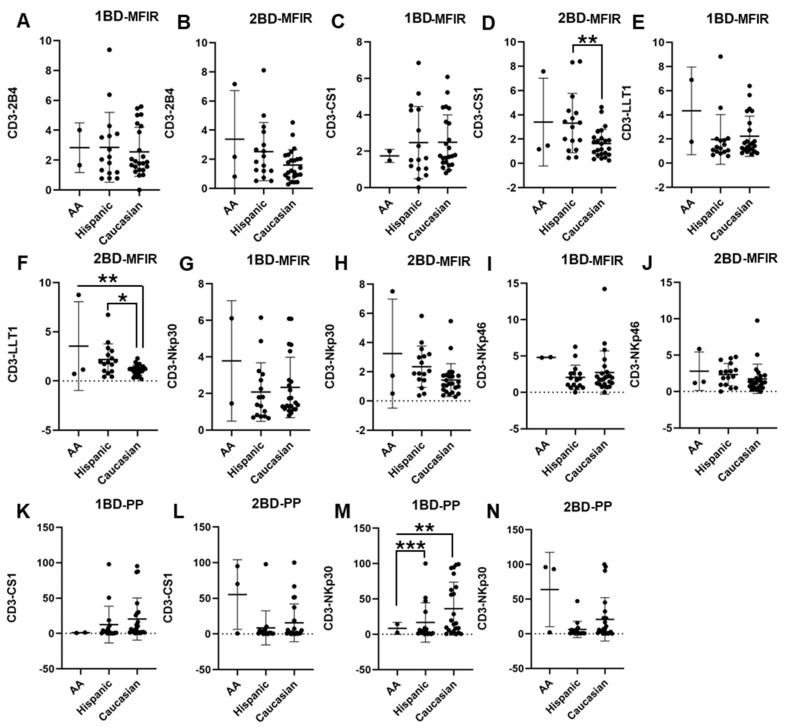
Differential expression of immune receptors on T cells of Hispanic, Caucasian, and African American ALL subjects before and after chemotherapy treatment. Flow cytometry analysis exhibits the cell surface expressions of 2B4 (**A**,**B**), CS1 (**C**,**D**), LLT1 (**E**,**F**), NKp30 (**G**,**H**), and NKp46 (**I**,**J**) on T cells of Hispanic, Caucasian, and African American ALL subjects pre- and post-chemotherapy treatments, as indicated by median fluorescence intensity ratio (MFIR). Flow cytometry analysis shows the percentage of positive (PP) cells expressing CS1 (**K**,**L**) and NKp30 (**M**,**N**) on T cells. The first blood draw is referred to as 1BD, which is collected at diagnosis or before the start of chemotherapy (day 0). The second blood draw is referred to as 2BD, which is collected at the end of induction chemotherapy or after treatment (day 29). * *p* < 0.05; ** *p* < 0.01; *** *p* < 0.001.

**Figure 2 biomedicines-13-01412-f002:**
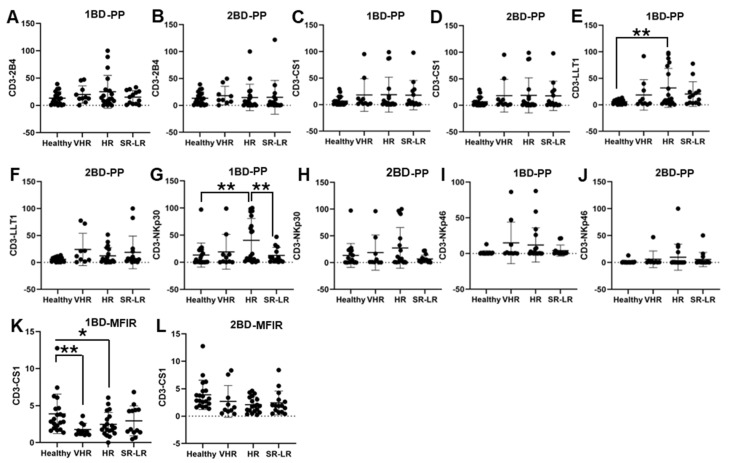
Cell surface expression of 2B4, CS1, LLT1, NKp30, and NKp46 on CD3+ T cells in high-risk ALL patients before and after chemotherapy treatment. Flow cytometry analysis of cell surface expression shows the percentage of positive (PP) cells expressing 2B4 (**A**,**B**), CS1 (**C**,**D**), LLT1 (**E**,**F**), NKp30 (**G**,**H**), NKp46 (**I**,**J**), and CS1 (**K**,**L**) indicated by median fluorescence intensity ratio (MFIR) on the CD3+ T cells of ALL subjects before and after chemotherapy treatments, stratified by risk-of-relapse. The first blood draw is referred to as 1BD, which is collected at diagnosis or before the start of chemotherapy (day 0). The second blood draw is referred to as 2BD, which is collected at the end of induction chemotherapy or after treatment (day 29). * *p* < 0.05, ** *p* < 0.01.

**Figure 3 biomedicines-13-01412-f003:**
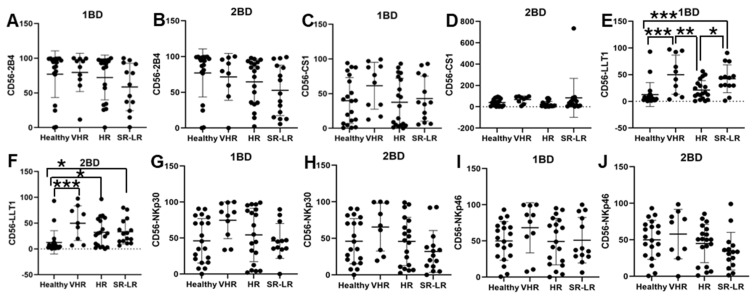
Cell surface expression of 2B4, CS1, LLT1, NKp30, and NKp46 on CD56+ NK cells of high-risk ALL subjects before and after chemotherapy treatments. Flow cytometry analysis of cell surface expression indicated by percentage of positive cells expressing 2B4 (**A**,**B**), CS1 (**C**,**D**), LLT1 (**E**,**F**), NKp30 (**G**,**H**), and NKp46 (**I**,**J**) on NK cells of ALL subjects before and after chemotherapy treatments stratified by risk-of-relapse. The first blood draw is referred to as 1BD, which is collected at diagnosis or before the start of chemotherapy (day 0). The second blood draw is referred to as 2BD, which is collected at the end of induction chemotherapy or after treatment (day 29). * *p* < 0.05, ** *p* < 0.01; *** *p* < 0.001.

**Figure 4 biomedicines-13-01412-f004:**
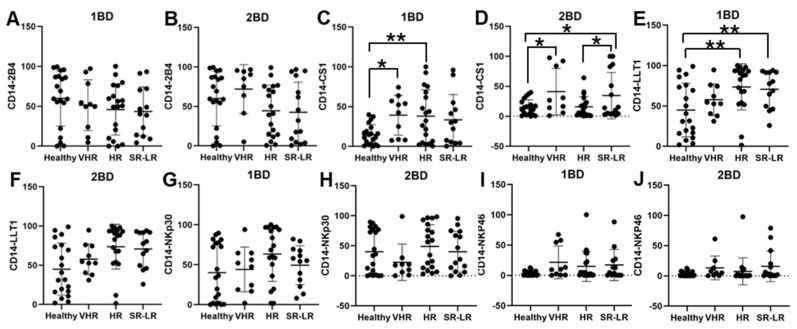
Cell surface expression of 2B4, CS1, LLT1, NKp30, and NKp46 on CD14+ monocytes of high-risk ALL subjects before and after chemotherapy treatments. Flow cytometry analysis of cell surface expression indicated by percentage of positive cells expressing 2B4 (**A**,**B**), CS1 (**C**,**D**), LLT1 (**E**,**F**), NKp30 (**G**,**H**), and NKp46 (**I**,**J**) on monocytes of ALL subjects before and after chemotherapy treatments, stratified by risk-of-relapse. The first blood draw is referred to as 1BD, which is collected at diagnosis or before the start of chemotherapy (day 0). The second blood draw is referred to as 2BD, which is collected at the end of induction chemotherapy or after treatment (day 29). * *p* < 0.05, ** *p* < 0.01.

**Table 1 biomedicines-13-01412-t001:** Demographic and clinicopathological features of ALL patients.

Clinical Features	No. of Patients
*Sex*	
Male	21
Female	21
*ALL Subtype*	
B-ALL	39
T-ALL	3
*CNS status*	
1 (no blasts in the CSF, regardless of WBC or RBC)	30
2a (<5 WBC/µL + blasts + <10 RBC/mL)	6
2b (<5 WBC/µL + blasts + ≥10 RBC/mL)	2
2c (≥5 WBC/µL + blasts + ≥10 RBC/mL)	2
3a (≥5 WBC/µL + blasts + <10 RBC/mL)	2
*MRD*	
Positive	6
Negative	34
Unknown	2
*DNA index*	
1.0	11
1.05–2.0	10
>2.0	9
Unknown	12
*Hyperdiploidy*	
Yes	12
No	19
Unknown	11
*Hypodiploidy*	
Yes	1
No	31
Unknown	10
*TEL/AML*	
Yes	2
No	40
*Philadelphia chromosome*	
Yes	6
No	36

## Data Availability

The data that support the findings of this study are available on request from the corresponding author, S.O.M. The data are not publicly available due to the information contained that could compromise the privacy of research participants.

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
