# Peer review of "Altered Expression of NK Receptors in Racially/Ethnically Diverse and Risk-of-Relapse Pediatric Acute Lymphoblastic Leukemia Patients"

_biomedicines, 2025, doi:10.3390/biomedicines13061412_

Round 1
Reviewer 1 Report
Comments and Suggestions for Authors
This is a scientifically valuable study that addresses a topical and very important topic, especially given the increasing role of immunophenotypic features in the prognosis and treatment of hematological diseases. However, to improve the article, I have provided a series of comments and suggestions that aim to clarify the findings, strengthen the mechanistic interpretations, and also highlight the clinical applicability of the results,
1-Determine whether the healthy groups match the patient group regarding age, sex, and race distribution. To eliminate bias, the match between groups must be examined and reported.
2-Determine whether the sampling time from the healthy group coincided with the time of 1BD/2BD.
3-Has the quality and purity of PBMC been checked after isolation? (e.g., by cell counting or viability assay)
4-An interesting point is made in the Discussion section about the interaction between LLT1 and NKRP1A. It is suggested that the discussion on the role of LLT1 in inducing T-cell exhaustion be further expanded, especially in the context of CAR-T therapy. Also, whether LLT1 can be a biomarker for predicting response to CAR-T.
5-Also, in the Discussion, although the role of NK receptors such as LLT1, CS1, NKp30, and NKp46 is mentioned, the specific cellular or molecular mechanisms that lead to the inhibition or activation of NK cells are not sufficiently described. It is suggested that this section be more prominent in the Discussion.
Author Response
This is a scientifically valuable study that addresses a topical and very important topic, especially given the increasing role of immunophenotypic features in the prognosis and treatment of hematological diseases. However, to improve the article, I have provided a series of comments and suggestions that aim to clarify the findings, strengthen the mechanistic interpretations, and also highlight the clinical applicability of the results,
1-Determine whether the healthy groups match the patient group regarding age, sex, and race distribution. To eliminate bias, the match between groups must be examined and reported.
Response: We thank the reviewer for the critical review of this manuscript. We want to state that as far as possible we attempted to match the healthy groups with the patient groups regarding age, sex and race/ethnicity distribution but we do acknowledge the fact that the match was not perfect due to the complexity of recruiting healthy children into a research study. This has been explicitly stated in the revised manuscript in section 3.5. The study underwent stringent regulatory review by the two IRB’s and despite the challenges and disadvantages of recruiting subjects from a single site we were able to recruit the number of patients reported in the study.
2-Determine whether the sampling time from the healthy group coincided with the time of 1BD/2BD.
Response: We want to confirm as stated in the Materials and Methods section that healthy subjects were recruited at the same time ALL subjects were recruited for the 1st blood draw (pre-chemotherapy treatment) and the whole blood was collected at the same time for the two groups. Another blood sample was collected only from the patients and not the healthy subjects on day 29 at the end of chemotherapy treatment, which was indicated as 2BD (2nd blood draw).
3-Has the quality and purity of PBMC been checked after isolation? (e.g., by cell counting or viability assay).
Response: We want to confirm that after PBMC isolation for every subject the viability of the cells was validated by counting the viable cells by cell counter and the purity of the isolated cells were determined by flow cytometry. 95% of the PBMC’s were viable cells and the purity of the PBMC samples were determined to be 98% pure.
4-An interesting point is made in the Discussion section about the interaction between LLT1 and NKRP1A. It is suggested that the discussion on the role of LLT1 in inducing T-cell exhaustion be further expanded, especially in the context of CAR-T therapy. Also, whether LLT1 can be a biomarker for predicting response to CAR-T.
Response: Per the reviewer’s recommendation, we have expanded the discussion on the role of LLT1 in inducing T-cell exhaustion. Please see discussion section.
5-Also, in the Discussion, although the role of NK receptors such as LLT1, CS1, NKp30, and NKp46 is mentioned, the specific cellular or molecular mechanisms that lead to the inhibition or activation of NK cells are not sufficiently described. It is suggested that this section be more prominent in the Discussion.
Response: We appreciate the reviewer for identifying this deficiency. We have included a paragraph in the beginning of the discussion section that discusses the molecular mechanisms that lead to the inhibition or activation of NK cells.
Reviewer 2 Report
Comments and Suggestions for Authors
This manuscript presents an insightful investigation into the expression of immune cell receptors among pediatric patients with Acute Lymphoblastic Leukemia (ALL), focusing on racial/ethnic disparities and the implications for relapse risk. The findings contribute to our understanding of immune dysregulation in ALL and have potential implications for personalized treatment strategies.
While the study includes a reasonable number of pediatric ALL subjects, the overall sample size of 42 patients may be considered limited, particularly when stratifying by race/ethnicity and risk of relapse. A larger sample size would enhance the statistical power and the generalizability of the findings.
Specific Comments:
- Abstract: The abstract succinctly summarizes the key findings; however, it could benefit from a clearer statement of the study's objectives and its significance in the context of current literature.
- Introduction: The introduction effectively sets the stage for the study, but it would be helpful to include recent references that discuss the role of NK cells in ALL and the implications of immune dysregulation in cancer relapse.
- Results Section: Consider providing additional visual aids (figures/tables) to summarize key findings for easier interpretation, especially for complex data sets.
- Discussion: comment further on how reacial disparities may impact chemoresistance based non different SNPs
- Conclusion: The conclusion could be strengthened by clearly outlining potential future directions for research, as well as the implications of this study for clinical practice.
Minor:
- Ensure consistent terminology throughout the manuscript (e.g., "very-high risk" vs. "high-risk" vs. "standard-risk/low-risk").
- Proofread for grammatical errors and clarity, particularly in sections with complex scientific terminology.
Author Response
Reviewer comments:
This manuscript presents an insightful investigation into the expression of immune cell receptors among pediatric patients with Acute Lymphoblastic Leukemia (ALL), focusing on racial/ethnic disparities and the implications for relapse risk. The findings contribute to our understanding of immune dysregulation in ALL and have potential implications for personalized treatment strategies.
While the study includes a reasonable number of pediatric ALL subjects, the overall sample size of 42 patients may be considered limited, particularly when stratifying by race/ethnicity and risk of relapse. A larger sample size would enhance the statistical power and the generalizability of the findings.
Response: We appreciate and thank the reviewer for the comments and do acknowledge the limitations of the small sample size. Globally, as compared to other cancers, ALL is relatively a cancer with low incidences. The current study was an exploratory research study conducted at a single center site which hampered our ability to enroll more subjects into the study. A future prospective study involving multiple centers would allow a more comprehensive analysis of ALL subtypes with a larger sample size. Due to this limitation our analysis did not focus on comparisons between the ALL subtypes: B-cell precursor versus T-cell ALL rather we pooled the samples and analyzed receptor expression pre-chemotherapy and post-chemotherapy treatments. There are no studies that have analyzed the risk factors or the racial/ethnic disparities of expression of these receptors in ALL and therefore we chose to analyze these five receptors for this study.
Specific Comments:
- Abstract: The abstract succinctly summarizes the key findings; however, it could benefit from a clearer statement of the study's objectives and its significance in the context of current literature.
Response: In light of the reviewer’s comment, we have revised the abstract to present a clear statement about the study’s objectives.
- Introduction: The introduction effectively sets the stage for the study, but it would be helpful to include recent references that discuss the role of NK cells in ALL and the implications of immune dysregulation in cancer relapse.
Response: Per the recommendations of the reviewer, we have added more recent references in the introduction section.
- Results Section: Consider providing additional visual aids (figures/tables) to summarize key findings for easier interpretation, especially for complex data sets.
Response: We have added two supplementary figures to better summarize key findings for easier interpretation, especially for complex data sets by explaining our gating strategy for the flow cytometry experiments.
- Discussion: comment further on how racial disparities may impact chemoresistance based on different SNPs
Response: Per the comments of the reviewer, we have extensively discussed how racial disparities may impact chemoresistance based on different SNPs and added more references starting at line 303. This has significantly improved the discussion on this topic.
- Conclusion: The conclusion could be strengthened by clearly outlining potential future directions for research, as well as the implications of this study for clinical practice.
Response: We have strengthened the conclusions by clearly outlining potential future directions for research, as well as the implications of this study for clinical practice.
Minor:
- Ensure consistent terminology throughout the manuscript (e.g., "very-high risk" vs. "high-risk" vs. "standard-risk/low-risk").
Response: We have thoroughly checked the manuscript and corrected all inconsistent terminologies.
- Proofread for grammatical errors and clarity, particularly in sections with complex scientific terminology.
Response: We have thoroughly checked the manuscript and rectified all grammatical errors especially in sections with complex scientific terminology.